# Forest Decline Triggered by Phloem Parasitism-Related Biotic Factors in Aleppo Pine (*Pinus halepensis*)

**Luna Morcillo** [1,*] , **Diego Gallego** [2,3] , **Eudaldo González** [4] **and Alberto Vilagrosa** [1,2,*]

1   Mediterranean Center for Environmental Studies (CEAM Foundation), Joint Research Unit University of Alicante-CEAM, Ctra. Sant Vicent del Raspeig s/n, Sant Vicent del Raspeig, 03690 Alicante, Spain
2   Departament of Ecology, University of Alicante, Ctra. Sant Vicent del Raspeig s/n, Sant Vicent del Raspeig, 03690 Alicante, Spain
3   Sanidad Agrícola Econex S.L., Calle Mayor 15Bis, Siscar-Santomera, 30149 Murcia, Spain
4   SILCO S.L, Calle Escalinata 12B, Guadarrama, 28440 Madrid, Spain
*   Correspondence: luna_morcillo@hotmail.com (L.M.); a.vilagrosa@ua.es (A.V.)

**Abstract:** Climate models predict increasing mean temperatures and reduced precipitation for Mediterranean ecosystems already subjected to major hydrological fluctuations. Forest decline phenomena relate extreme droughts or heat waves with other organisms, e.g., insects or microorganisms acting as pests, but their role needs to be elucidated. A biotic factor responsible for forest diseases is *Candidatus Phytoplasma pini* which is a phloem-parasitism that negatively affects Spanish pine forests in drought-prone areas. In several healthy and declining Aleppo pine stands, we monitored pine infection by PCR (Polimerase Chain Reation), determined the tree phloem tissue terpene composition, carbohydrate content, measured several relevant morpho-physiological variables and examined trees affected by bark beetles. PCR confirmed *C. P. pini* infection was widespread in all stands, regardless of to the presence of symptomatically affected trees. However, visible symptomatic decline only occurred in trees living under more stressful conditions. The terpene composition of pines in declining stands differed from those in healthy ones, and could be related with bark beetle attacks when pines were previously weakened by the phytoplasma disease. Our results indicate that biotic factors, such as *C. P. pini*, affecting phloem tissue may be triggering factors for drought-mediated forest decline and suggest that phloem diseases can play a key role in forest declining processes during extreme drought.

**Keywords:** climate change; drought; forest decline; phloem disease; phytoplasma; tree growth; tree mortality; triggering factors

## 1. Introduction

Forests are extremely valuable ecosystems given the many services they provide. However, increasing temperatures and fewer annual rainfall events, as predicted by the most recent climate models, will generate hotter drier environments that will severely affect forest ecosystems [1]. Increased aridity is particularly expected in Mediterranean regions [2,3], which might entail more frequent drought recurrence and the risk of forest pests and diseases [4–6]. In fact, recent changes in temperature and rainfall regimes related to global change, termed "hotter droughts", seem to promote decaying processes and mortality in several forest ecosystems worldwide [6–8]. As a result, not only forests' species composition, but also their ecological functioning and hydrological properties, such as water balance or infiltration for aquifers recharge, will be severely affected [9].

Some theories have been proposed to explain certain ongoing mechanisms underlying plant mortality observed in recent years [10,11]. However, the physiological mechanisms that underlie drought mortality are still not fully understood, and may be related to severe reductions in carbon stocks in some cases (i.e., decreasing non-structural carbohydrate pools), and hydraulic failure due to extreme drought in others [12,13]. Other authors have suggested that these two processes can be complementary and not exclusive [10,14,15]. In addition to these causes, forest decline processes can be exacerbated by interactions with other organisms (i.e., pest attacks), such as insects or some microorganisms [5,16–18] which, among other consequences, may alter the carbohydrate reserves used by trees to produce defense chemicals [19,20]. Indeed, considerations about the role of phloem in transport and plant defenses enhance the present relevance of these scenarios [21,22].

Forest diseases, such as insects, fungi or bacteria outbreaks, are often preceded by drought events, which stress host trees that become more vulnerable to such attacks [5,23,24]. Temperature can also influence insects through development and survival rates to determine population success [25,26]. Other features, like tree abundance, density, size and physiology, and their spatial pattern distribution, have also been described to directly affect insect populations' capacity to grow and spread [23]. Consequently, different stress factors that affect trees can reduce pines' resin production by affecting the resin canal network or impairing resin exudation, leading to more intense and dangerous attacks [11,27]. This may entail the disappearance of or significant reduction in some species' distribution areas if they are unable to adapt or migrate fast enough while favoring other species at the same time [28]. For all these reasons, understanding the causes and being able to predict the future of our forests is crucial to determine the best management practices to mitigate the ecological consequences of ongoing climate change.

Phytoplasmas are mycoplasma-like bacteria devoid of cell walls. This specialized microorganism is a strict parasite of the phloem tissue, and seems to be vectorized by homopteran insects of the superfamilies Cicadellidae, Fulgoridae and the family Psyllidae [29,30]. They have been described as disease-causing in about 1000 plant species, mainly angiosperms [31]. Phytoplasmas may entail host plant metabolic changes, such as disrupted hormonal balance, impaired amino acid and carbohydrate translocation, photosynthesis inhibition and rapid senescence [32–34]. Infected plants show a wide range of symptoms and, particularly in coniferous forest trees, infection can cause uncontrolled branching proliferation in specific branches (commonly known as witches' brooms), but also defoliated shoots, dwarfed needles and stunted growth [35]. Schneider and others [36] have associated these symptoms in Scots pine (*Pinus sylvestris* L.) and Aleppo pine (*Pinus halepensis* Miller) trees in Germany and Spain with infection caused by the taxon '*Candidatus Phytoplasma pini*' (hereafter referred *C. P. pini*), identified by the Polimerase Chain Reation (PCR) amplification of 16S rDNA and sequence analyses. These results have been corroborated in other conifer species (*Pinus banksiana* Lamb., *Pinus mugo* Turra, *Pinus nigra* J.F.Arnold, *Pinus tabuliformis* Carr., *Abies procera* Rehder and *Tsuga Canadensis* (L.) Carrière) by some authors in Poland, Lithuania and the Czech Republic [35,37], in that this microorganism is also detected in non-symptomatic trees, evidencing a non-direct relation between phytoplasma and disease [36]. Despite the existence of a few studies confirming the association among several anomalous growths in pines with *C. P. pini* infection-related symptoms, the *C. P. pini* life cycle and transmission between trees of remains still unknown due to the impossibility of isolation and consequently no studies on vectors or acquisition mechanisms are available. Moreover, very little is known about its effects on tree mortality, morpho-physiological performance, carbohydrate reserves and induced defense compounds, all of which are considered key factors related to forest decay processes that face expected climate change scenarios.

In the present study, we assess the effects of *C. P. pini* forest infection by selecting several Aleppo pine stands in a gradient of affected trees with different degrees of disease: (i) healthy stands (i.e., only with non-symptomatic trees); and (ii) declining stands with non-symptomatic and symptomatic trees (i.e., stands containing trees with visible and non-visible symptoms). In all the stands, we analyzed *C. P. pini* infection by PCR methodology, and measured phloem tree terpene composition as an indicator of

tree-induced defense processes. These measures were completed by analyzing several functional traits related to morphological and ecophysiological variables and carbohydrate content to establish the impact of disease on pine vigor and performance. We also analyzed the mortality rate in these stands and the impact of bark beetles (Coleoptera: Curculionidae, Scolytinae) on the mortality processes of affected trees.

We hypothesize that *C. P. pini* will differently affect tree performance for all response parameters according to the degree of symptoms, with lower performance in symptomatic trees followed by the non-symptomatic trees coexisting within declining stands and better performance in healthy stands.

## 2. Materials and Methods

### 2.1. Experimental Set-Up

Based on previous observations made by the Regional Forest Services, which pointed out some scattered mortality events some years ago in several Aleppo pine populations located in central Spain, in June 2017 we selected several declining and non-declining stands in two study areas, Cerros Concejiles (40°16′47.18′′ N; 3°28′54.43′′ W) and Cerro Palomero (40°5′0.09′′ N; 3°22′37.95′′ W) forests. Both areas are the result of old Aleppo pine plantations implemented during the 1970s. Reforestations were carried out with pines coming from the same nursery and seed source, as the Regional Forest Service reported to us.

The climate in both areas is Mediterranean-temperate characterized by cold winters and intense summer droughts, with an average annual precipitation and temperature of 455 mm and 14 °C respectively (1951–1999 data period from the Arganda weather station, 40°18′23.16′′ N; 3°26′52.43′′ W, Spanish Meteorological Agency, AEMET), however, the climatic information for the 10 years prior to the experiment shows an increasing trend in the average temperature and a decreasing trend for annual precipitation (Figure 1).

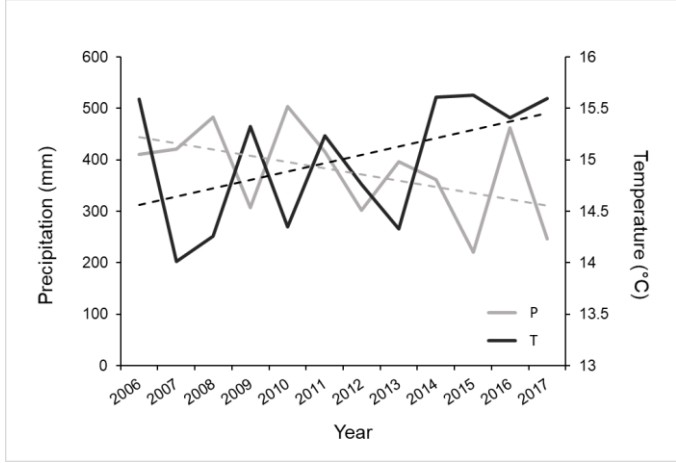

**Figure 1.** Annual precipitation (in continuous grey line) and average temperature (in continuous black line) for the 10 years previous to the experiment. Dashed lines show the tendency through time for both parameters.

The area of study are smooth hills with basic soils that come from marly limestone sedimentary rocks. The soil of these areas consists of loamy soils, Gypsic cambisol according to the FAO soil classification. The main traits are low fertilization, high carbonate contents and a basic pH (Table 1). The dominant understory vegetation is dominated by alpha-grass tussocks (*Stipa tenacissima* L.) associated with other perennial grasses, such as *Brachypodium retusum* (Pers.) P.Beauv. 1812, and evergreen shrubs, and subshrubs like *Rhamnus lycioides* L., *Quercus coccifera* L., *Rosmarinus officinalis* L., Sp. Pl.,1, 23, 1753 or *Thymus vulgaris* L.

**Table 1.** Main soil characteristics in the area. OM: Organic matter, N: nitrogen, P: phosphorous, K: potassium, Fe: iron.

| Parameters | Mean ± Error |
|---|---|
| OM (%) | 0.81 ± 0.44 |
| N (%) | 0.19 ± 0.00 |
| P (ppm) | 13.2 ± 1.9 |
| K (ppm) | 22.4 ± 8.5 |
| Fe (%) | 0.63 ± 0.18 |
| Carbonates (%) | 42.6 ± 5.9 |
| Stoniness (%) | 30.2 ± 5.1 |
| Texture: | |
| Sand (%) | 32.4 ± 2.7 |
| Silt + Clay (%) | 68.7 ± 3.2 |

The different degrees of disease in each tree and stand were established by visual criteria: healthy stands with non-symptomatic pines (NA-trees) where no declining processes or pine mortality was observed, and declining stands (i.e., stands with decaying pines and tree mortality) within which it was possible to distinguish between non-symptomatic pines with no evident disease signs (NS-trees), and symptomatic pines with visible disease symptoms (S-trees). Symptoms were related to dry leaves on apparently healthy branches, mortality of branches and twigs, defoliation degree and needle malformations due to abnormal growth in terminal shoots (see Figure S1 in Supplementary Materials for descriptive pictures).

We selected a total of five forest stands: two healthy stands, Concejiles and Palomero, (C and P, respectively) and three declining stands: Concejiles-1 (C1), Concejiles-2 (C2) and Palomero-1 (P1). Six pines displaying each symptomatology type in all the stands were chosen, which resulted in 12 NA pines, 18 NS and 18 S pines. Thus, the total amount of pine trees considered herein was 48.

*2.2. Measurements*

2.2.1. The *C. P. pini* Infection PCR Test

The presence, not the amount, of infection of *C. P. pini* in pine individuals was tested at the beginning of the experiment, by sampling actively growing twigs in all 48 selected trees. In order to prevent any cross-contamination of samples, shoots were manually broken without using cutting instruments, and gloves were discarded after sampling each tree. PCR identifications were made in a specialized laboratory (Dr. Cobos's Lab, Polytechnic University of Madrid, Madrid, Spain). Following the protocol of Schneider and others [36], DNA extraction was performed with an initial enrichment step, followed by a procedure that employed the NucleoSpin Plant II kit (Macherey-Nagel). The primers used for the nested-PCRs were the phytoplasma-specific ribosomal primer pairs fU5/rU3 and P1/P7. A re-amplification of the P1/P7 product with internal primers R16F2/R2 was also performed to obtain 880 bp fragments from the 16S rRNA gene [36]. Positive PCR products were visually tested by electrophoresis in agarose gels [37].

2.2.2. Induced Defense Compounds

To assess the induced defense of the trees undergoing a declining process, we analyzed the phloem tissue terpene composition. Three trunk core samples from three pines per stand (C, C1 and C2) and disease degree (NA, NS and S) were collected in Cerros Concejiles in July 2017. This sampling was carried out at the same time than the first sampling of ecophysiological traits. After discarding some samples that were unable to be analyzed, the final number was three samples from NA trees, five samples from S trees and four samples from NS trees. We used a Pressler's auger to take two opposite cores at the same level from the ground in each tree. After removing the bark, phloem and cambium were separated from sapwood. Using a scalpel, tissue was sealed into a headspace 15 mL vial to be

quickly frozen in liquid nitrogen to stop any subsequent biosynthesis of new terpenoid compounds. Then, the samples were stored at −25 °C for further laboratory analyses.

The relative terpenoid composition was obtained by following a modified protocol of Kelsey et al., [38] with a headspace analysis using a 65-μm PDMS/DVB (Polydimethylsiloxane/Divinylbenzene) fiber (HS-SPME, Solid Phase Microextraction in Head Space) in a GERSTEL MPS 2XL autosampler attached to an Agilent GC 7890B gas chromatograph coupled to a 5977A mass spectrometer (GC-MS). Vials were incubated for 5 min at 50 °C, and the fiber was exposed in a headspace for 1 min. The injection temperature was set at 220 °C by maintaining the fiber desorption pendant for 1 min. The column used was a HP-5MS 5% Phenyl Methyl Silox (30 m × 250 μm × 0.25 μm). The column oven was held at 40 °C for 1 min and was then increased to 220 °C at 5 °C/min. Helium was the carrier gas, with 7.07 psi and a constant column flow of 1 mL/min throughout the run. The terpenoid compounds were identified by mass spectra comparison in the NIST 11 mass spectral database with a retention time. The absolute chromatographic area was used as an indirect measure of the relative terpenoid content without standards, according to the applied protocol [38].

### 2.2.3. Mortality Rates

Mortality rates were estimated once at the end of the experiment, by counting all the dead individuals in a specific area. We were unable to estimate recent mortality in Palomero-1 because selective thinning works to remove dead trees had been carried out in the area. Only the trees that had recently died (i.e., pines that died less than two years ago) were recorded according to the visual parameters. To determine if the mortality period was less than two years, we considered if trees had brown or gray needles attached to twigs and if there were any signs of bark detachment, according to previous experiences [17]. We also determined the possible causes of pine death by assessing the evidences of bark beetle attacks (*Tomicus destruens* and *Orthotomicus erosus*). Moreover, we considered the presence of pitch tubes and exit holes on the trunks, with their species identification by subcortical gallery observations.

### 2.2.4. Morphological Traits

Several morphological traits related to tree performance were evaluated at the end of the growing period, early July. The leaf mass fraction (LMF) was estimated as the fraction between the needle dry biomass and the total dry biomass (i.e., needle plus wood). We also measured the relative presence of dry needles (i.e., dry biomass in relation to total needle biomass) and the fraction between the number of active buds and the branch section. The tree trunk diameter at breast height (DBH) was also recorded at the beginning of the study.

### 2.2.5. Ecophysiological Traits

Plant water status was evaluated by measuring the shoot-water potential at predawn and at midday in a Schölander pressure chamber (Model 1000, Pressure Chamber, Instrument, PMS Instrument Company, Albany, OR, USA). Samples were taken and placed inside a dark bag until measurements were taken, which took no longer than 15 min after cutting. Leaf photosynthetic performance as the maximal PSII (Photosystem II) photochemical efficiency (Fv/Fm) was determined with a portable PAM-2100 fluorometer (Heinz Walz GmbH, Effeltrich, Germany). Fv/Fm was measured mid-morning in the samples pre-darkened for 30 min after which the maximal PSII photochemical efficiency (Fv/Fm) was recorded according to standard protocols [39]. All the samples were determined in the same trees previously used for the morphological determinations twice in this study: T1, before mid-summer (July) and T2, after summer 2017 (September).

In September, coinciding with the second sampling of ecophysiological traits, the non-structural carbohydrates concentration (NSC) was estimated as starch and soluble sugar contents (SS; including sucrose, glucose and fructose) in the twigs of four trees per stand and disease degree. Samples were frozen in liquid nitrogen in a Dewar container in the field to then be transferred to a freezer where were

stored at −25 °C for further analyses in a specialized laboratory (Agrolab Analítica SL, Pamplona, Spain) where they were oven-dried at 75 °C and ground in an electrical mill to pass a 1-mm screen. Starch was determined by the enzymatic method (amyloglucosidase and glucose-oxidase–peroxi-dase) and was then measured colorimetrically in a visible spectrophotometer at 500 nm UV wavelength. Soluble sugars (total and reductors) were extracted by reduced ferrocyanide, and their concentration was determined colorimetrically in a visible spectrophotometer at 540 nm UV wavelength. These samples were taken after the summer of 2017 to register the minimum values after the adverse summer period.

2.2.6. Data Analysis

We analyzed the terpenoid composition data by running k-means clustering and creating a hierarchical clustering tree (dendrogram) by the hclust method. We estimated the optimum number of clusters according to the gap curve. To evaluate the differences in terpene content between disease degrees, we ran a permutational multivariate analysis of variance (PERMANOVA). To assess the differences between disease degrees in the amount of the six major terpene compounds, we performed an analysis of variance (ANOVA) for the unbalanced data (type III). Tukey's HSD (Honestly Significant Difference) tests were used for the mean pair comparisons. The statistical analyses for terpenoids were performed with R [40] by the vegan, car and cluster libraries.

To evaluate the differences between symptomatic affected pines (S-trees) and the unaffected non-symptomatic pines (NS-trees) within the declining affected stands for the morphological and ecophysiological variables, we ran a two-way ANOVA with stand and disease degree as the fixed factors. This analysis was carried out for each survey separately (t1 and t2) when applicable. A post hoc test was also performed to evaluate the differences among pine stands for all the variables. To compare between the degrees of effects (NA and NS trees), we used an analysis of variance with degree of effect as a fixed factor for each survey separately (t1 and t2). All data meet the normal distribution of residuals and homoscedasticity assumptions. Statistical analyses were performed by using the v.23.0 Statistical package (SPSS Inc., Chicago, IL, USA).

## 3. Results

### 3.1. Analyzing C. P. pini Infection

The PCRs markers were positive for *C. P. pini* in all samples in the tests carried out in agarose gels. Detection of this phytoplasma in the entire sample set implied infection, independently of the observed degree of disease (NA, NS or S-trees).

### 3.2. Induced Defense Compounds

The content of terpenoids determined in phloem tissues consisted of 133 compounds (including mono- and sesquiterpene, and their oxygenate products), of which 76 substances were identified by their mass spectra. The clustering analysis of the k-means separated the samples into two differentiated groups or clades (Figure 2). The biggest group included all the samples from the declining stands, independently of the symptomatology (i.e., NS-trees and S-trees). The other group comprised all the pine samples from the healthy stands (NA-trees). Most of the samples from the decaying stands (NS and S pines) were arranged as three minor clades, with no apparent relation with disease degree.

The relative abundance of the secondary compounds in the constitutive phloem showed significant differences among the disease degrees (Figure 3). The sesquiterpene isocaryophyllene were significantly more abundant in the trees of the healthy stands (NA-trees) than in those of declining stands (NS-trees and S-trees). Contrarily, the monoterpenes camphene, D-limonene, and an unknown monoterpene, and sesquiterpenes caryophyllene and humulene were differentially found in the trees of declining stands.

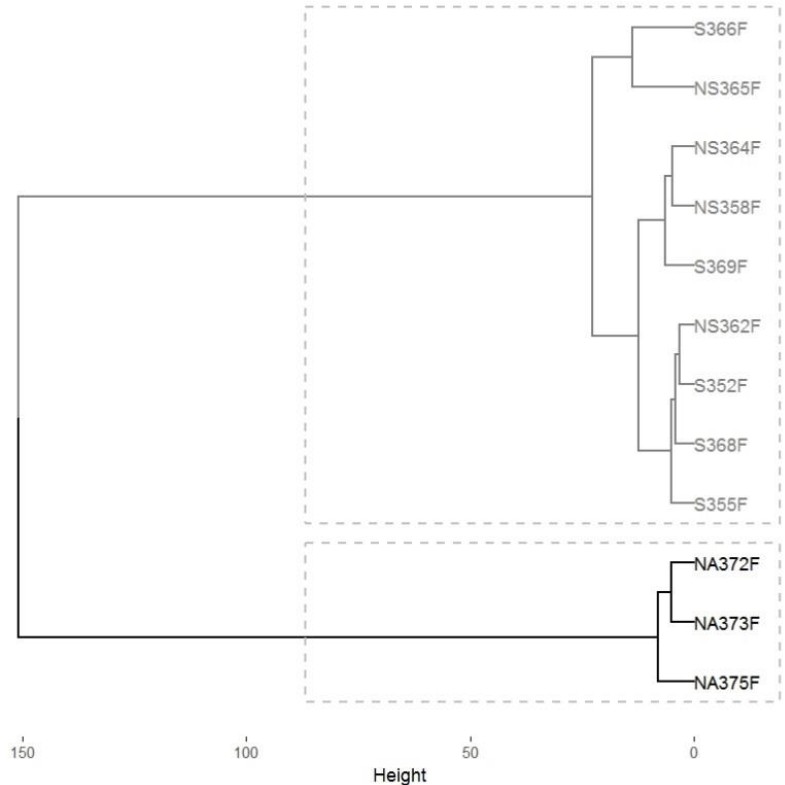

**Figure 2.** K-means cluster dendrogram for the terpenoid composition in the phloem–cambium tissue samples. Dotted-line rectangles indicate the k-means clustering groups. Numbers correspond to the pine sample identification and letters before each code represent the disease degree (NA: non-symptomatic pines in healthy stands, NS: non-symptomatic pines within declining stands, S: symptomatic pines within declining stands).

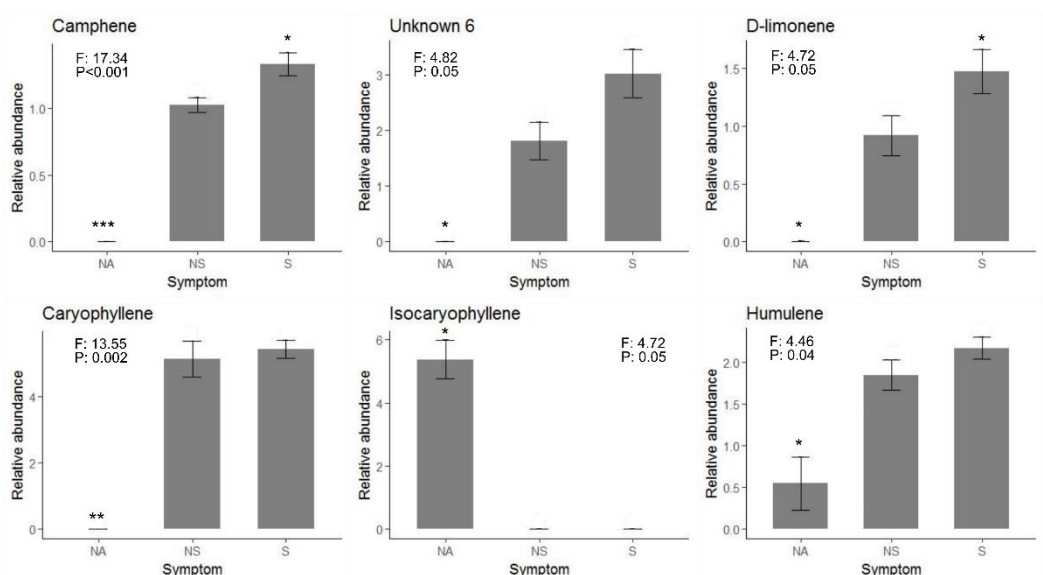

**Figure 3.** Relative abundance (%) of the six main secondary compounds for disease degrees (NA: non-symptomatic pines in healthy stands, NS: non-symptomatic pines within declining stands, S: symptomatic pines within declining stands). Data are mean ± SE of $n = 5$ pines per degree of disease. Asterisks represent differences between disease degrees: $p < 0.001$ (***), $p < 0.01$ (**) and $p < 0.05$ (*).

### 3.3. Morphological Traits

Comparing healthy and declining stands, the diameter at breast height (DBH) showed no differences between NA-trees and NS-trees (Figure 4, Table 1). Within declining stands the DBH indicated smaller diameters in S-trees than in NS-trees (Figure 3, Table 2).

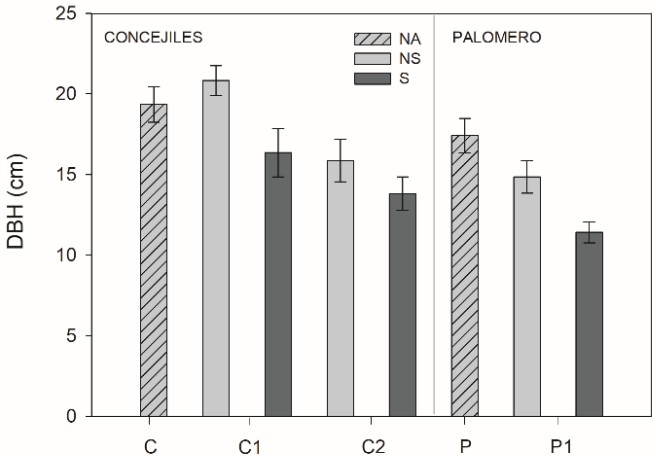

**Figure 4.** Diameter at breast height (DBH, cm) in each pine stand and disease degree. The striped dark gray bars show the values for the NA-trees; the smooth light gray bars show the values for the NS-trees; the smooth dark gray bars show the values for the S-trees. Data are mean ± SE of $n = 6$ pines per degree of disease and stand.

**Table 2.** Summary statistics of Linear Model for the measured variables for disease degrees (NA and NS). Significant differences at $p < 0.05$ are shown in bold.

| Variables | Disease Degree F (P) |
|:---:|:---:|
| DBH | 0.93 (0.343) |
| LMF | 0.47 (0.500) |
| Terminal buds | 0.00 (0.948) |
| Dry needles | 0.93 (0.343) |
| $WP_{pd}$ | $T_1$: **13.91 (0.001)** <br> $T_2$: **9.89 (0.004)** |
| Fv/Fm | $T_1$: 3.37 (0.078) <br> $T_2$: **11.01 (0.003)** |
| Starch | **7.35 (0.015)** |
| Soluble sugars | **4.66 (0.046)** |
| Total NSC | **5.66 (0.030)** |

Leaf mass fraction (LMF) showed higher needle biomass for non-symptomatic pines (NA and NS-trees) than for symptomatic ones (S-trees) (Figure 5, Table 3). No differences between the NA and NS-trees were found for this variable (Figure 5, Table 2).

The relation between the number of terminal buds per branch section indicated no differences between NA and NS-trees. However, we found a smaller amount of buds in S-trees for all stands (Figure 5, Table 3).

The proportion of dry needles per branch was higher in the symptomatic individuals (S-trees) than in non-symptomatic pines (NA-trees and NS-trees) (Figure 6, Table 3). The maximum values were between 10%–20% in relation to the total amount of needles growing in the terminal twigs.

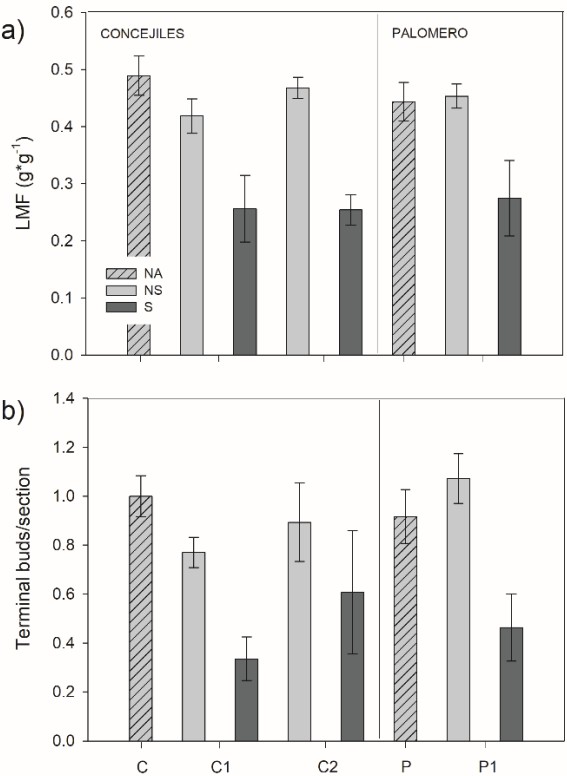

**Figure 5.** Panel (**a**) Leaf mass fraction ($g \times g^{-1}$) measured for each pine stand and disease degree. The striped dark gray bars show the values for the NA-trees; the smooth light gray bars show the values for the NS-trees; the smooth dark gray bars show the values for the S-trees. Data are mean ± SE of n = 6 pines per stand and disease degree. Panel (**b**) Relation between the number of terminal buds by branch section ($mm^{-2}$), measured for each pine stand and disease degree. The striped dark gray bars show the values for the NA-trees; the smooth light gray bars show the values for the NS-trees; the smooth dark gray bars show the values for the S-trees. Data are mean ± SE of *n* = 6 pines per stand and disease degree.

**Table 3.** Summary statistics of Linear Model for disease degree (NS and S) at different pine stands (C1, C2, and P1). Significant differences at *p* < 0.05 are shown in bold and the marginal significant differences in italics. Acronym "nd" represents no differences (i.e., *p* > 0.05) in the post hoc test between stands, "S" refers to stand factor and "D" refers to disease degree factor.

| Variables | Stand (S) F (P) | Disease Degree (D) F (P) | S × D F (P) | *Post hoc* Stand |
|---|---|---|---|---|
| DBH | 12.80 (<0.001) | **13.60 (0.001)** | 0.61 (0.549) | P1 = C2 ≤ C1 |
| LMF | 0.25 (0.779) | **30.00 (<0.001)** | 0.20 (0.821) | nd |
| Terminal buds | 1.31 (0.284) | **13.62 (0.001)** | 0.60 (0.554) | nd |
| Dry needles | 1.12 (0.340) | **6.21 (0.018)** | 0.85 (0.439) | nd |
| $WP_{pd}$ | $T_1$: **5.09 (0.012)**<br>$T_2$: **10.71 (<0.001)** | 0.52 (0.477)<br>0.05 (0.824) | 0.58 (0.567)<br>1.84 (0.190) | C2 ≤ P1 ≤ C1<br>C2 ≤ P1 ≤C1 |
| Fv/Fm | $T_1$: **12.54 (<0.001)**<br>$T_2$: *3.03 (0.076)* | 0.85 (0.362)<br>0.39 (0.540) | 0.03 (0.970)<br>0.87 (0.438) | C2 < C1, P1<br>nd |
| Starch | *3.04 (0.076)* | 2.47 (0.136) | 0.64 (0.542) | nd |
| Soluble sugars | **4.88 (0.022)** | **18.69 (0.001)** | 2.02 (0.166) | nd |
| Total NSC | **5.73 (0.013)** | **19.18 (<0.001)** | 2.20 (0.143) | C1 ≤ P1≤ C2 |

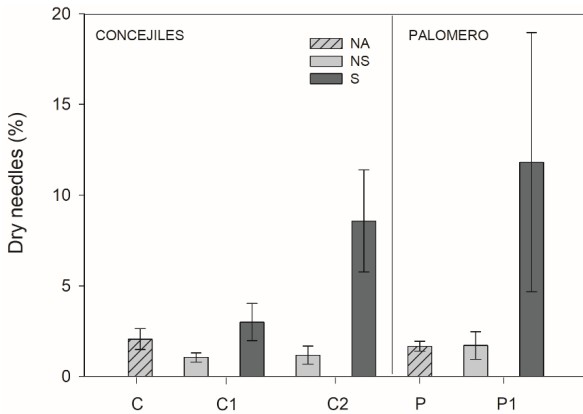

**Figure 6.** Proportion of dry needles versus normal needles per branch (%), measured in each pine stand and disease degree. The striped dark gray bars show the values for the NA-trees; the smooth light gray bars show the values for the NS-trees; the smooth dark gray bars show the values for the S-trees. Values are the mean ± SE for *n* = 6 pines per stand and disease degree.

### 3.4. Plant Water Status and PSII Functionality

The water potential measured at predawn (Ψpd) showed that trees in the unaffected stands (NA-trees) had had better water status conditions, i.e., higher (less negative) water potential values, than the non-symptomatic individuals in the decaying stands (NS-trees, Figure 7). This finding reflects that the pines in the healthy stands had better water status conditions (Table 2). The comparison between NS-trees and S-trees within declining stands showed no significant differences, and the differences were only found at stand level (C1 compared to C2) (Table 3).

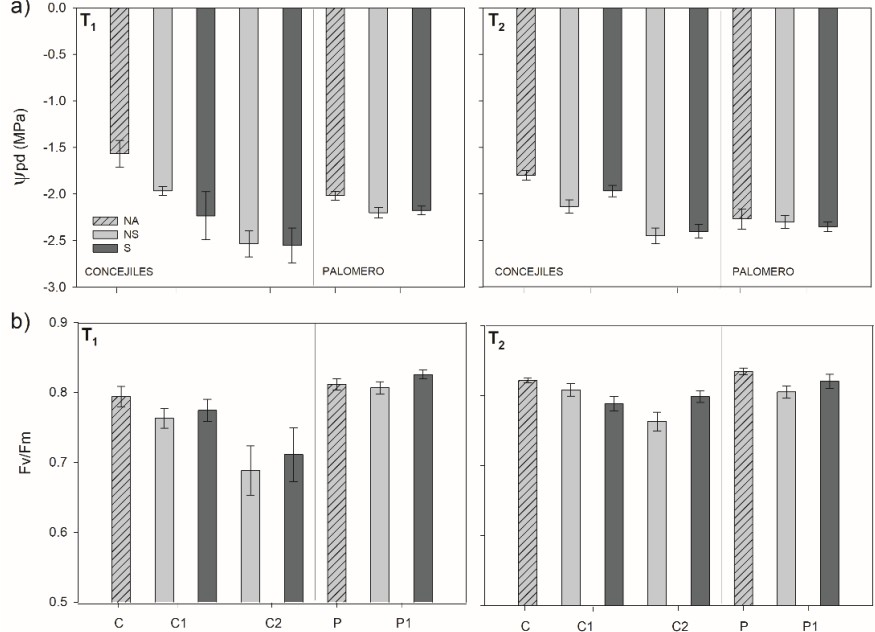

**Figure 7.** Panel (**a**) Water potentials measured at predawn (Ψpd) measured in each pine stand and disease degree, for both surveys T1 and T2. The striped dark gray bars show the values for the NA-trees; the smooth light gray bars show the values for the NS-trees; the smooth dark gray bars show the values for the S-tees. Values are the mean ± SE for *n* = 6 pines per stand and disease degree. Panel (**b**) Maximum PSII photochemical efficiency (Fv/Fm), measured in each pine stand and disease degree, for both surveys T1 and T2. The striped dark gray bars show the values for the NA-trees; the smooth light gray bars show the values for the NS-trees; the smooth dark gray bars show the values for the S-tees. Values are the mean ± SE for *n* = 6 pines per stand and disease degree.

The maximum photochemical PSII efficiency (Fv/Fm) followed the same trend. The non-symptomatic individuals in healthy stands (NA-trees) obtained higher values for this variable (around 0.8) than the non-symptomatic individuals in decaying stands (NS-trees) (Figure 7, Table 2). Within declining stands, NS and S-trees did not show significant differences between disease degrees (Table 3). The results indicated that C2 had lower Fv/Fm values than C1 and P1 during the first survey (T1), with no differences in T2 (Table 3). Regarding the sampling date, we observed only a slight increase in declining stands.

### 3.5. Carbohydrate Concentration

The concentration of the total non-structural carbohydrates within declining stands was higher in NS-trees than in S-trees (Figure 8, Table 3). The same results were found for the soluble sugar fraction (SS). However, no significant differences between NS and S-trees were found for starch content (Figure 8, Table 3). The post hoc test showed significant differences among stands for total NSC (Table 3). We also found significant differences between NA-trees and NS-trees for all the fractions (Figure 8, Table 2).

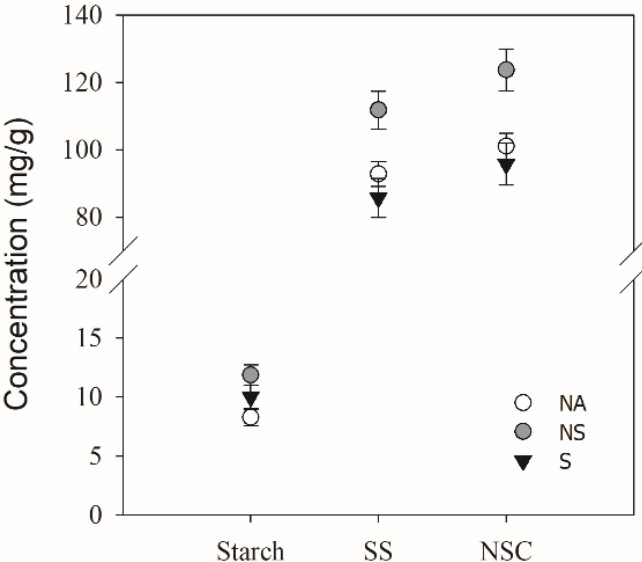

**Figure 8.** Concentration of starch, soluble sugars and total NSC for each disease degree. Empty circles represent the NA mean values, gray circles represent the NS mean values and black triangles represent the mean S values. Values are the mean ± SE for *n* = 4 pines per stand and disease degree.

### 3.6. Pine Mortality and Bark Beetle Attacks

The number of dead pines varied among pine stands (Figure 9). We did not find dead pines in the Concejiles or Palomero stands, considered as healthy stands. The mortality rates recorded in the declining stands C1 and C2 were about two to four pines/ha during the last two years.

The presence of bark beetles indicated significant differences between stands. In C1 all dead trees presented attacks by these insects, while only 70% of the dead trees were affected by bark beetles in C2. Attacks by *Tomicus destruens* were more marked at C1, while a bigger portion of dead trees in C2 showed evidence for a mixed attack with *Orthotomicus erosus*. In contrast, the proportion of attacks by *O. erosus* was similar at both sites.

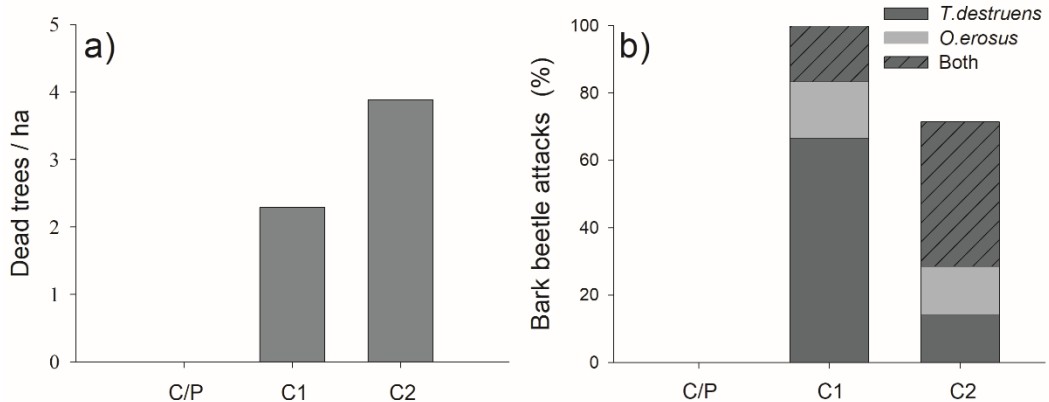

**Figure 9.** Panel **a**) Tree mortality (dead trees/ha) in each stand considering only the trees that had recently died (i.e., trees that died less than two years ago). Panel **b**) Percentage of dead trees evidencing bark beetle attacks in each pine stand. The smooth dark gray stack represents attacks by *Tomicus destruens*; the smooth light gray stack represents attacks by *Orthotomicus erosus*; the striped dark gray stack represents mixed attacks by both species. Recent logging did not enable us to determine the mortality and bark beetle attacks in P1.

## 4. Discussion

Pine morphological and growth anomalies as the observed in the present study are related to *Candidatus Phytoplasma pini* disease according PCR analysis and observations in previous studies [30,35–37]. However, the generalized presence of this microorganism in apparently healthy trees points out that the disease symptomatology must be related with other biotic or abiotic triggering factors. According to our results, drought stress conditions entailed more negative impacts of this disease on the pine populations, causing the decline of these populations. In fact, the pluviometry records in the area has been decreasing during the last ten years and, at the same time, annual mean temperatures increased. Under these conditions, the level of drought stress through soil water availability conditions or atmospheric demand has increased resulting in harsher conditions for some of these stands.

### 4.1. Phytoplasma Infection and Induced Defense Mechanisms

The presence of *C. P. pini* was detected in all analyzed samples, which supports the wide distribution of this infection at the forest scale as described in some previous studies [41]. However, visual tree decline signs related to phytoplasma compatible symptomatology were only observed in some stands. In fact, the presence of this phytoplasma in non-symptomatic trees has also been reported by several authors [35–37] who pointed out an unusual pathological expression of *C. P. pini* [36]. Therefore, two phenomena occur in the studied area. First, a general presence of *C. P. pini* in healthy and unhealthy trees, and secondly a differential expression of pine disease (phytoplasma symptom compatible), which could be related to other factors such as climatic conditions or a differential degree of water stress affecting pines in the stands.

Terpenoid compounds are commonly found in nature acting as semi-chemicals of feeding deterrents, fungicides or defensive compounds [19]. Since *C. P. pini* is a parasite of the phloem, changes in terpenoid composition must be also expected as a reaction to pathogenic infection [42]. Variations in these compounds as a response to biotic attacks have been largely reported [43–47]. For instance, Achotegui-Castells et al., [48] described a positive relation between the concentration of terpenoids in *Cupressus sempervivens* infected by *Seiridium cardinale*, which causes the progressive infection known as cypress canker. Accordingly, in our case, the terpenoid composition in phloem differed depending on the health condition of the stands. Regardless of their symptoms, the pines from declining stands (NS-trees or S-trees) displayed a different composition than pines from healthy stands (NA-trees). Indeed, quantitative terpenoids (monoterpene and sesquiterpene types) formed a clear grouping pattern of trees in two separated clades, related with disease prevalence instead of *C.P.*

*pini* presence. The same pattern was found in the relative abundance of the eight major secondary compounds described. While the relative abundances of the monoterpenes (-)-α-pinene, D-limonene, camphene and one unknown compound, and the sesquiterpenes caryophyllene and humulene, were higher in the trees from declining stands (S and NS trees), isomer (+)-α-pinene and sesquiterpene isocaryophyllene were more abundant in trees from healthy stands (NA trees). These differences in the pattern of expression could constitute the initially induced response in those trees (NS trees) that have not evidenced declining symptoms yet. All trees are from the same nursery and seed source, as the Regional Forest Service reported to us, so different pine chemotypes are unlikely. On the other hand, future quantification of *C. P. pini* in host tree tissues by rt-PCR or Q-PCR could clarify if the induced level of response is correlated with microorganism concentrations.

### 4.2. Effects of C.P. pini Infection on the Morphological and Eco-Physiological Functional Traits

The effect of *C. P. pini* symptoms was remarkable in some specific morphological traits measured. Although all the studied stands were simultaneously out-planted around the 1970s, the NS-pines living in declining stands had larger DBHs than the S-trees living in the same stands. The results at branch level showed a similar trend, with smaller needle biomasses for symptomatic pines (S-trees) than for non-symptomatic ones (NS-trees). The negative effect of these symptoms on S-trees was also evident in the higher percentage of dry needles and active terminal buds. Our findings agree with other studies that reported slower growing and atrophied needles in trees with abnormal shoot branching as a result of phytoplasma infection in *P. halepensis*, and other naturally infected species such as *Pinus banksiana*, *Pinus mugo*, *Pinus nigra*, *Pinus tabuliformis*, *Abies procera* and *Tsuga Canadensis* [35–37,49].

Contrarily to morphology, the physiological status between the symptomatic (S-trees) and non-symptomatic trees (NS-trees) living in the same stands did not differ for any sampling period. However, the pines living in healthy stands (NA-trees) displayed a better water status (less negative water potential) and greater photochemical efficiency than those living in the declining stands. Therefore, these results pointed out that the stands with better water conditions would favor healthier pine conditions by avoiding the occurrence of declining processes, possibly related to more intense drought conditions. Despite the water potential values recorded in our study reflecting mild stress conditions for Mediterranean species, the water potential for S and NS pines reflected intense stress conditions, since Aleppo pine is an isohydric species with a stomatal closure of about −2.5 MPa [50].

Although pathogens can cause direct tree death through the production of toxic metabolites, they can also induce functional damage by hydraulic failure, photosynthetic inhibition or carbon starvation by altering NSC demand or supply. The relationship between pathogen infection and the physiological mechanisms of drought-induced mortality is complex and depends on the trophic interaction established with the tree [51]. In our study, SS and the total NSC contents in twigs of the pines living in declining stands were higher in NS-trees than in S-trees, while no differences were found for starch. This finding agrees with other works conducted in declining Scots pine populations, where some authors found lower NSC values in defoliated pines than in non-defoliated pines [52,53]. Other experimental studies in conifer saplings [54] and pine species [55] have found drastic reductions in NSC (especially starch) in stems upon mortality, but have not indicated completely depleted carbohydrate storage [56]. In fact, under similar conditions to those of our study, García de la Serrana et al., [17] observed that pines were still able to retain some level of NSC, even when they were in an extremely declined state. Thus, the activation of different biosynthetic routes would require a proportional increment in the metabolism of NSC reserves and carbon allocation by consequently reducing carbohydrate reserves. Our study evidenced that monoterpene and sesquiterpene acted as part of the active primary defense compounds in our populations, which is energetically expensive to build and maintain. Thus, the activation of different biosynthetic routes would require a proportional increment in the metabolism of NSC reserves consequently reducing carbohydrate reserves [20,57]. Surprisingly, the values for all the fractions were also higher in NS-trees than in NA-trees. In a recent study conducted with two *Pinus* spp. (*P. contorta* and *P. Ponderosa*), Piper et al., [58] provided evidence

for carbohydrate accumulation (i.e., a long-term precautionary strategy in starch and short-term osmoregulation in soluble sugars) and for growth decline in response to drought. This could explain our results, as NS-trees live according to a worse water status than the unaffected pines in symptomatic stands and are, therefore, submitted to more stressful conditions than NA-trees, which would incite these NS-trees to store NSC to be alert.

### 4.3. Bark Beetle Infection and Mortality Associated to C. P pini Disease

As a result of the decline process in our Aleppo pine populations, tree mortality and associated bark beetle attacks were observed in declining stands, while no signs were detected in healthy stands. Forest ecologists have long recognized that stressed trees, e.g., those affected by drought, are more vulnerable to insect attack [59,60]. In coniferous trees attacked by bark beetles, the induced reaction of resin impregnation of the tissues surrounding the point of aggression plays a determining role in the tree's resistance [59,61]. Since induced resin production is costly in energy terms [62], the resistance against possible attacks would depend on the tree's vigor at the time of attack [63,64]. We noticed the absence of pitch tubes in *T. destruens* attacks, which evidenced that attacked pines were unable to generate defensive resin fluxes. This suggests that *P. C. pini* progressive infection based on disease expression signs could be the main factor that triggered mortality, and that the bark beetles implied in tree mortality acted as opportunists by attacking trees in an advanced state of decline. Phloem is an important aspect for plant defenses and its action could be blocked or minimized due to drought conditions [22,27,51]. In this sense, drought and phytoplasma disease could act synergistically as phloem sieves infected tubes, which reduces phloem transport capacity and would decrease pine capacity to respond to beetle attacks.

## 5. Conclusions

This study confirms the widely extended phytoplasma *C. P. pini* infection in all pine populations analyzed and also shows strong evidence that phytoplasma symptoms play a major role in the decline of Aleppo pines populations. Our findings show that not only drought-induced declining processes operate in our forests. However, the visible phytoplasma symptoms and the decline processes associated in the trees only appeared under more stressful conditions, reflecting the synergistic effect of the lack of water and biotic diseases. Since both phytoplasma disease expression and drought act as stressors by affecting the plant's capacity to provide a plastic response, we may expect their combined occurrence would have additive negative effects and could lead more rapidly to exhaustion. This would result in a more accelerated decline process. However, the mechanisms that lie behind *C. P. pini* symptoms in forests are still poorly known and more studies are needed to better understand how *C. P. pini* acts pathologically and the level of microorganism concentration to reach a threshold for the symptomatic expression.

In terms of management applications, we suggest promoting straightforward actions for forest management of drought–pathogen interactions to prevent as much as possible both water stress in trees during drought episodes and pathogen control measures to avoid phytoplasma from spreading to healthy individuals and stands. In this sense, pine forests resulting from past reforestations in dry to semiarid climates conditions beyond optimal ranges could be more vulnerable to "hotter droughts" mediated by biological damaging agents.

**Supplementary Materials:** The following are available online at http://www.mdpi.com/1999-4907/10/8/608/s1, Figure S1: (A) General view of affected pine forests in Cerros Concejiles (Madrid, Spain), with affected pines in the first term. (B) Tree crown detail with defoliated branches. (C) Initial stage of defoliation with dry leaves. (D) Morphological alterations of twig growth and leaves development. (E) Recently dead tree which retain brown leaves. (F) A percentage of dead trees were affected by bark beetles (*Tomicus destruens* and *Orthotomicus erosus*) with high amount of sub-cortical galleries surrounding all the trunk and holes where young beetles have gone.

**Author Contributions:** Conceptualization, L.M., D.G., E.G. and A.V.; methodology L.M., D.G., E.G. and A.V.; software, L.M. and D.G.; validation, L.M., D.G. and A.V.; formal analysis, L.M., DG and A.V.; investigation, L.M., D.G. and A.V.; resources, D.G., E.G. and A.V.; data curation, L.M., D.G., E.G. and A.V.; writing—original draft preparation, L.M. and D.G.; writing—review and editing, L.M., D.G. and A.V.; supervision, A.V.; project administration, A.V.; funding acquisition, E.G. and A.V.

**Funding:** This research was funded by Sección de Defensa Fitosanitaria (Conservación del Medio Natural, Consejeria de Medio Ambiente, Comunidad de Madrid) and the Survive-2 project (CGL2015-69773-C2-2-P MINECO/FEDER) from the Spanish Government. The CEAM Foundation is supported by Generalitat Valenciana.

**Acknowledgments:** We thank Pablo Cobos Lab from the Polytechnic University of Madrid for its assistance in the molecular identification of samples.

**Conflicts of Interest:** The authors declare no conflict of interest. The funders had no role in the design of the study; in the collection, analyses, or interpretation of data; in the writing of the manuscript, or in the decision to publish the results.

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
