# Peer review of "Forest Decline Triggered by Phloem Parasitism-Related Biotic Factors in Aleppo Pine (Pinus halepensis)"

_forests, doi:10.3390/f10080608_

Round 1
Reviewer 1 Report
Comments for the authors
Overall, the paper is interesting and comprehensive. Phytoplasmas are difficult to identify through macroscopically and the reasons for their development are not well known and understood. Therefore, any information on Phytoplasmas are very useful. However, there are some points that should be improved in order to make the paper more informative. There are two major points that should be clarified before the publication of the paper:
(a) you should explain your arguments for drought stress on the examined trees. Differences on the water status of the trees do not necessarily indicate drought stress. Have you gathered information on the rainfall of the area? Please see also the comments below. In case your work is based on data (of drought stress) from previous research you should provide this information to the readers.
(b) You provide some general information on the soil type of the examined areas but you did not collect soil samples for a soil analysis; such an analysis would allow you to explain better possible ecophysiological symptoms on the trees. You claim that all the differences are observed due to the presence of Phytoplasmas whereas the development of Phytoplasmas may be the just the response of the tree to other factors (see aslo comments below).
Please see also the comments bellow.
Comments
General
Ø It would be helpful and informative to describe better the disease cycle of Candidatus phytoplasma pini and provide. When are the trees more susceptible? what about the dissemination? etc
Ø You don’t give an estimate of the amount of water/rainfall for the season. How did you conclude that the trees are under drought stress without having an estimate of the rainfall for the season?
Ø How many times did you sampled from the trees? I believe that sampling within time would have been more informative. Sampling e.g. for two successive years would provide more information on the disease dissemination and severity and would allow you to acquire information under different amounts of rainfall.
Ø The reasons for the development of Phytoplasmas are not well known. However, often extreme temperatures between winter and summer time may also trigger their development. Have you checked the temperatures for the year that you sampled the trees?
Ø What about the role of mycorrhizas? There are certain species that are thought to help trees cope with drought stress. I believe that you should have sampled roots from the examined sites and trees and examine them for mycorrhiza presence. Mycorrhizas play an important role on the ecophysiology of the trees and is important to be aware of their existence in order to assess their resistance to the disease under drought stress.
Ø Statistical analysis and presentation with letters of the different groups is a bit old fashioned, although acceptable. You should also mention (in Materials and Methods section) which statistical package you used for the analysis. It would be also informative if you could explain why you chose this type of analysis-factorial ANOVA-instead of REML or other analysis which are more suitable for such type of data. You should also explain why you analysed separately each survey and how many shoots did you sampled per tree.
Ø Please correct in italics all the species names throughout the text (e.g. line 75- use italics for P. sylvestris, lines 108-109 etc)
Author Response
Reviewer #1 Comments for the authors
Overall, the paper is interesting and comprehensive. Phytoplasmas are difficult to identify through macroscopically and the reasons for their development are not well known and understood. Therefore, any information on Phytoplasmas are very useful. However, there are some points that should be improved in order to make the paper more informative. There are two major points that should be clarified before the publication of the paper:
(a) you should explain your arguments for drought stress on the examined trees. Differences on the water status of the trees do not necessarily indicate drought stress. Have you gathered information on the rainfall of the area? Please see also the comments below.
In case your work is based on data (of drought stress) from previous research you should provide this information to the readers.
Reply: Thank you for your appreciations. We revised the ms. to improve and clarify our statements about your comments. Our study is supported by previous analysis of drought stress in plants. One of the corresponding authors has a large experience in the analysis of drought impact on plant species, from leaf level to vulnerability to cavitation, see for example Vilagrosa et al., 2003; Vilagrosa et al., 2010; Vilagrosa et al., 2014; García de la Serrana et al., 2015; among others). Specilfically for P. halepensis, water potential at stomatal closure is estimated about -2.5 MPa (Baquedano & Castillo, 2006 and 2007) and xylem embolism levels start from -3.5MPa (García de la Serrana et al., 2015). In our study water potentials at the beginning of the summer were from -1.5 in NA stands to -2.5 MPa in the driest C2 affected stands at predawn as it is shown in Fig 6. These values were maintained along the time until the second sampling carried out after summer (September). According to these values, pines were under moderate to intense levels of drought stress if we take into account that these values were registered at predawn, that suppose lower values than at midday.
We have also reviewed the data about rainfall regime in the area (see comments below) and we made improvements in the text to be more readable.
We have done changes in several paragraphs in the ms. in the Material & Methods and Results sections to clarify our statements (see detailed comments below).
(b) You provide some general information on the soil type of the examined areas but you did not collect soil samples for a soil analysis; such an analysis would allow you to explain better possible ecophysiological symptoms on the trees. You claim that all the differences are observed due to the presence of Phytoplasmas whereas the development of Phytoplasmas may be the just the response of the tree to other factors (see aslo comments below).
Reply: We agree with the referee´s comments in the importance of soil characteristics on ecophysiological symptoms on the trees, but soils in the area are quite homogeneous (according to the comments of the Forest Service and our own appreciations) and consequently, soil analysis were not within our objectives. In fact, the area is composed by smooth hills with basic soils come from marly limestone sedimentary rocks. However, we consider that this is an interesting point and we have included some more information about the soils in the area taken from a previous study (L 120-L 124 and Table 1). We hope the reader can more clearly understand the soil type in the area of study.
Reviewer #1_Specific Comments
General
Ø It would be helpful and informative to describe better the disease cycle of Candidatus phytoplasma pini and provide. When are the trees more susceptible? what about the dissemination? Etc
Reply: Thanks for this comment, accordingly we have reviewed the introduction and discussion sections and included some paragraphs giving more information about this disease and the host trees (L74-L84 in the Introduction and L365-370 Discussion section). However, due to the impossibility of C.P. pini isolation, caused by the lack of cell walls, the disease cycle of this pathogen has not been examined and it is not available in the literature, (L80 - L82).
Ø You don’t give an estimate of the amount of water/rainfall for the season. How did you conclude that the trees are under drought stress without having an estimate of the rainfall for the season?
Reply: Thank you very much for this useful suggestion. Following it, we have incorporated climatic information for the previous 10 years before the experiment (L111-L115 and Fig 1). That information shows an increasing trend in the average temperature and a decreasing trend for annual precipitation, which may have supposed a cumulative abiotic stress in the studied forests. This observation has also been included in the Discussion section (L366-370).
Ø How many times did you sampled from the trees? I believe that sampling within time would have been more informative. Sampling e.g. for two successive years would provide more information on the disease dissemination and severity and would allow you to acquire information under different amounts of rainfall.
Reply: These suggestions probably have resulted from a lack of information in the previous version of the manuscript. We have included a better description of sampling dates: The presence of C. P. pini in pine individuals was tested at the beginning of the experiment (L134). Accordingly, Mortality rates were estimated once at the end of the experiment (L186), Several morphological traits related to tree performance were evaluated at the end of the growing period, early July (L197-202), and Ecophysiological traits were determined in the same trees previously used for the morphological determinations twice in this study: T1, before mid-summer (July) and T2, after summer 2017 (September) (L204-214). The samples for non-structural carbohydrates were taken after the summer of 2017 to register the minimum values after the adverse summer period (L214-219). We fully agree with the reviewer that sampling for successive years would offer more information about the disease, unfortunately, it was not possible due to duration of the project and funding reasons. However, the samplings were carried out twice in the same year, analyzing the period before mid-summer and after summer under higher accumulated water stress.
Ø The reasons for the development of Phytoplasmas are not well known. However, often extreme temperatures between winter and summer time may also trigger their development. Have you checked the temperatures for the year that you sampled the trees?
Reply: As happen with the C.P. pini cycle or disease transmission, there are no studies about the impact of temperature or lack of precipitation on the spread of this disease. In our study, the average wintry temperature between January and February was 6.4°C that compared to the average summer period, between July and August that was 26.3°C, results in a difference of 19.9°C between winter and summer 2017. No differences with the records taken previous years were observed since the difference between the minimum temperature and the maximum showed an average about 19.1ºC (range: 17.7-22.2ºC) for the last 12 years. Therefore, nothing seems to indicate this effect as a factor to take into account. We consider that we do not contribute too much to the knowledge gap by supplying this info to the manuscript. No comments have been included in the ms. since we do not consider this point as relevant.
Ø What about the role of mycorrhizas? There are certain species that are thought to help trees cope with drought stress. I believe that you should have sampled roots from the examined sites and trees and examine them for mycorrhiza presence. Mycorrhizas play an important role on the ecophysiology of the trees and is important to be aware of their existence in order to assess their resistance to the disease under drought stress.
Reply: We agree that mycorrhizas can play an important role for the trees. In our study we did not sampled for mycorrhizas analysis. However, the selected forests are conformed by adult stands and consequently we would not expect changes in mycorrhizas or fungi flora in general, apart from those associated to the microsite conditions.
Ø Statistical analysis and presentation with letters of the different groups is a bit old fashioned, although acceptable. You should also mention (in Materials and Methods section) which statistical package you used for the analysis. It would be also informative if you could explain why you chose this type of analysis-factorial ANOVA-instead of REML or other analysis which are more suitable for such type of data. You should also explain why you analysed separately each survey and how many shoots did you sampled per tree.
Reply: As suggested by the reviewer, the letters in figure 3 have been replaced by numeric F and p values and asterisks to show differences between disease degree (L269-L272).
We used the statistical tools appropriate to differentiate inequalities in population means. We consider that ANOVA and ANOVA type III (for unbalanced data) are suitable and widely used tests for these kind of comparisons. Mixed Models, as LMER (or REML as Referee suggested) is also a robust test, usable for more complex data structures than the used in our study, which random and fixed effects should be included.
We have included additional information about statistical packages in the proper sub-section within Materials and Methods (Data Analyses) in L233-234 and L242-244.
Ø Please correct in italics all the species names throughout the text (e.g. line 75- use italics for P. sylvestris, lines 108-109 etc)
Reply: We fixed the mistakes in italics throughout the manuscript.
Reviewer 2 Report
The manuscript titled “Forest decline triggered by phloem parasitism-related biotic factors in Aleppo pine (Pinus halepensis)” describes results that show tree physiology and defenses are impacted by infection by Candidatus Phytoplasma pini and that drought (i.e., local growth conditions) and development of symptoms can be linked to infestation by bark beetles, therefore resulting in a forest decline. The paper was clear, well organised and well written – I only have minor editorial points listed below. And I think that this manuscript is a novel contribution and will be well received because it integrates many measures to look a not well understood pathogen/forest decline.
My only major comment is about needing more details in the methodology. When were induced defenses and carbohydrates taken? Were these at the same time for all stands? And did they correspond to physiological measures? These compounds can change within a season and between years. Therefore, a little more detail on sampling timing would be helpful. Furthermore, more detail on the GC method is needed (Ln 148), even if in supplemental material. I also think that the flow rate needs to be checked, I think the maximum constant flow on a column with this dimensions is 2.0 mL/min for He. I’m surprised that the HP-5MS column was able to separate chiral compounds and the GC/MS method would help with this interpretation. Also, why were authentic standards not used for verification of MS results and does this alter the confidence in the identification? Was Kovats retention index used for the sesquiterpenes, as the NIST library can list many compounds for these classes of compounds?
Ln 14 – change to “heat waves”
Ln 15-16 – revise to “A biotic factor responsible for a forest disease is Candidatus Phytoplasma pini, which is a phloem-parasitism that negatively affects Spanish pine forests in drought-prone areas.”
Ln 18 – change word order to “tree phloem tissue terpene composition” here and throughout
Ln 19 – I’m unsure what “censored” means in this sense. Maybe revise to “examine” if this is what is meant.
Ln 23 – change beetles to beetle
Ln 75 – italicize P. sylvestris
Ln 79 – remove “trees’ at” and “on”
Ln 90 – correct to “(Coleoptera: Curculionidae, Scolytinae)”
Ln 92 – revise to “according to the”
Ln 106-109 – italicize scientific names of species, here and throughout MS and check Ln 315-316.
Ln 141 – seems redundant to say samples were frozen again in liquid N – maybe revise sentence to state just what they were stored at.
Ln 168 – revise to “presence of dry”
Ln 185 – revise to “where they were”
Ln 202 – I think there needs to be a space between declining and affected
Ln 208-209 – the sentence cuts off without finishing what program and packages were used to analyze the rest of the data – needs fixed.
Ln 225 – “Figure 1” is repeated in caption.
Ln 335 – revise to “semiochemicals to feeding”
Ln 350-352 – But what are other possibilities to explain these results? Have there been chemotypes identified in Aleppo pine? Or what is the origin of the seed for these plantations – could they be from different seed sources?
Ln 382-385 – The first sentence needs fixed as it abruptly stops without finishing and the second needs removed as it is a repeat.
Ln 398 – Is this what is meant “incite these trees to not store NSC.”
Ln 403 – revise to “coniferous trees attacked”
Tables and figures: Abbreviations used should be spelled out in the caption or notes.
Supplemental Material: Figure 1 should be relabeled as Figure 1A.
Author Response
Reviewer #2 Comments for the authors
The manuscript titled “Forest decline triggered by phloem parasitism-related biotic factors in Aleppo pine (Pinus halepensis)” describes results that show tree physiology and defenses are impacted by infection by Candidatus Phytoplasma pini and that drought (i.e., local growth conditions) and development of symptoms can be linked to infestation by bark beetles, therefore resulting in a forest decline. The paper was clear, well organised and well written – I only have minor editorial points listed below. And I think that this manuscript is a novel contribution and will be well received because it integrates many measures to look a not well understood pathogen/forest decline.
My only major comment is about needing more details in the methodology. When were induced defenses and carbohydrates taken? Were these at the same time for all stands? And did they correspond to physiological measures? These compounds can change within a season and between years. Therefore, a little more detail on sampling timing would be helpful.
Reply: Thanks to the reviewer for the positive and valuable comments, accordingly we carefully reviewed the methodology and make improvements in the description of the methods and sampling periods of. The samples to asses induced defenses were taken in July 2017, coinciding with the first sampling of ecophysiological traits (L162-164). The non-structural carbohydrates concentration (NSC) was estimated for all stands in September, coinciding with the second sampling of ecophysiological traits, (L-214)
Furthermore, more detail on the GC method is needed (Ln 148), even if in supplemental material. I also think that the flow rate needs to be checked, I think the maximum constant flow on a column with this dimensions is 2.0 mL/min for He. I’m surprised that the HP-5MS column was able to separate chiral compounds and the GC/MS method would help with this interpretation. Also, why were authentic standards not used for verification of MS results and does this alter the confidence in the identification? Was Kovats retention index used for the sesquiterpenes, as the NIST library can list many compounds for these classes of compounds?
Reply: Thank you very much for your comments, they allowed us to solve some mistakes regarding secondary defense compounds. We have reviewed our results and modified the text including several considerations made by the reviewer such as: 1) Flow rate has been fixed (changed from 54ml/min to 1 ml/min); 2) References to Chiral compounds have been removed as we agree with referee’s considerations and Figure 3 has been corrected in order to avoid any possible mistake about identification of compounds; 3) We would like to clarify that compound identification has been made based on MS spectrums of NIST library; 4) -We have cited the protocol used for terpenoid determination (L-181-184), which establish that “The absolute chromatographic area was used as an indirect measure of the relative terpenoid contents without standards”, according to the applied protocol proposed by Kelsey et al., 2014 [reference#38]”
.
Ln 14 – change to “heat waves”
Done
Ln 15-16 – revise to “A biotic factor responsible for a forest disease is Candidatus Phytoplasma pini, which is a phloem-parasitism that negatively affects Spanish pine forests in drought-prone areas.”
Reply: The reviewer’s suggestions have been incorporated
Ln 18 – change word order to “tree phloem tissue terpene composition” here and throughout
Reply: The reviewer’s suggestions have been incorporated
Ln 19 – I’m unsure what “censored” means in this sense. Maybe revise to “examine” if this is what is meant.
Reply: Yes, we referred to “control or examine”. We replace censured for examined, as suggested by the reviewer (L 20)
Ln 23 – change beetles to beetle
Reply: Done (modified in L 24)
Ln 75 – italicize P. sylvestris
Reply: Done (modified in L 72)
Ln 79 – remove “trees’ at” and “on”
Reply: Done (modified in L 83)
Ln 90 – correct to “(Coleoptera: Curculionidae, Scolytinae)”
Reply: Corrected in L-94
Ln 92 – revise to “according to the”
Reply: Done (Modified in L 93)
Ln 106-109 – italicize scientific names of species, here and throughout MS and check Ln 315-316.
Reply: Italics corrected in L 328-L329
Ln 141 – seems redundant to say samples were frozen again in liquid N – maybe revise sentence to state just what they were stored at.
Reply: The suggestion has been incorporated in L 172
Ln 168 – revise to “presence of dry”
Reply: Modified in L200 “the presence of dry needles”
Ln 185 – revise to “where they were”
Reply: Modified in L219
Ln 202 – I think there needs to be a space between declining and affected
Reply: Space added between declining and affected in L 237
Ln 208-209 – the sentence cuts off without finishing what program and packages were used to analyze the rest of the data – needs fixed.
Reply: Fixed in L 233-234
Ln 225 – “Figure 1” is repeated in caption.
Reply: Corrected
Ln 335 – revise to “semiochemicals to feeding”
Reply: Modified in L-387
Ln 350-352 – But what are other possibilities to explain these results? Have there been chemotypes identified in Aleppo pine? Or what is the origin of the seed for these plantations – could they be from different seed sources?
Reply: Paragraph related to possible chemotype diversity in the study area has been modified, including a sentence on the homogeneus origin of tree plants (L405-407, Discussion section) and L106-108 in Material and Methods section.
Ln 382-385 – The first sentence needs fixed as it abruptly stops without finishing and the second needs removed as it is a repeat.
Reply: Reviewed and fixed in L443-446
Ln 398 – Is this what is meant “incite these trees to not store NSC.”
Reply: Here we tried to explain the highest accumulation in NSC in non-symptomatic trees, compared to the NA trees, which apparently may result surprising. However, these NS trees living already within declining stands could be storing NSC to be alert. We replace the word “prevented” by “alert” and clarify that we refer to NS-tress in order to make the reasoning more understandable (L 460-462). We hope it is clear now.
Ln 403 – revise to “coniferous trees attacked”
Reply: Reviewed and fixed
Tables and figures: Abbreviations used should be spelled out in the caption or notes.
Reply: Abbreviations from the table have been remove to the table header as suggested
Supplemental Material: Figure 1 should be relabeled as Figure 1A.
Reply: Done
Reviewer 3 Report
Manuscript forests-538935 by Morcillo & al investigates the role of a phytoplasma, C. P. pini, in Aleppo Pine decline. This represents a difficult task that I do not at all underestimates, having worked myself on forest decline. A medium sample (48 tree in 5 stands) is characterized for many health (dbh, needle browning, nb active apex), physiological (terpene phloem content) and pathological (C. P. pini presence in phloem), which appears at first very nice. This manuscript however present some serious problems which make the conclusion claimed unsupported (“strong evidences that C. P. pini plays a major role in the decline of Aleppo pines”). The major problem of this work is that no “healthy” control is present. All the trees analyzed are infected with the studied Phytoplasma, whether healthy in stands without decline or healthy / declining in stands with decline. With this situation it becomes very difficult to reach any conclusion on the effect of C. P. pini in the studied phenomenon because no measure of infection severity is provided. For example, L335-352 it is finally concluded that the change in bark terpene content between NA – NS/S trees could be a initial sign of C. P. pini infection (visible in NS trees before they decline). How can that be while NA trees are similarly infected by C. P. pini? Throughout the paragraph this remains unclear. Terpene content may reflect infection difference, but what does differentially infect NA/NS/S trees? Throughout the discussion, this remains. The smaller dbh, higher %brown needle, nb terminal buds by branch section are “negative effect of C. P. pini on S-trees”. Ok, but nothing in the results presented support that, as all your trees you studied are infected by C. P. pini and no measure of infection severity is given. You might have another cryptic pathogen (for example in roots, or virus or whatever) that induce that and that you did not detect.
The fact that 100% of the studied trees are infected, whether they are in perfectly healthy stands or in heathy trees in stands with presence of decline is interesting and little commented. It is just said that it is expected that they are “widespread”. I must say that like most forest pathologist, I am not familiar with Phytoplasma. Is the 100% prevalence “normal”; this wouldn’t be unique, some root pathogens like some common Phytophthora or some leaf pathogens (oak mildew for example) might have a 100% prevalence in a sample of 5 stands. But then, there will likely be very large difference of severity between trees. Is that what is expected with Phytoplasma (difference in pathogen load in the phloem or something similar)? If so, to reach the conclusion you claim, you need to document those differences.
Some more minor comments:
L135-138. What is done is not easy to follow. 12 samples altogether and three pines per stand and disease degree. This should be 3 trees * 2 stands without symptoms +3 trees * 3 stands with symptoms *2 disease degree (healthy/declining) which would be 20 altogether. Can you clarify.
Fig. 8. I am not sur I have understood. I expected 4 stands (P, C, C1, C2), while in P1, logging precluded to determine mortality. P is not on the figure. Is that a mistake? Should it be P/C in the X-axis instead of C? Indicate in the caption that recent logging did not enabled to determine mortality in P1.
Author Response
Reviewer #3 Comments for the authors
Manuscript forests-538935 by Morcillo & al investigates the role of a phytoplasma, C. P. pini, in Aleppo Pine decline. This represents a difficult task that I do not at all underestimates, having worked myself on forest decline. A medium sample (48 tree in 5 stands) is characterized for many health (dbh, needle browning, nb active apex), physiological (terpene phloem content) and pathological (C. P. pini presence in phloem), which appears at first very nice.
This manuscript however present some serious problems which make the conclusion claimed unsupported (“strong evidences that C. P. pini plays a major role in the decline of Aleppo pines”). The major problem of this work is that no “healthy” control is present. All the trees analyzed are infected with the studied Phytoplasma, whether healthy in stands without decline or healthy / declining in stands with decline. With this situation it becomes very difficult to reach any conclusion on the effect of C. P. pini in the studied phenomenon because no measure of infection severity is provided.
Reply: We want to express that our research team is made up by forest pathologists with wide experience in forest diseases. In addition, we count with the additional support of the local Forest Services (Dept. of Environment, Comunidad de Madrid, Spain) which are conducting a long-term monitoring of the selected forest in this study. The healthy stands were selected, according to the Forest Service´s indications, since they were apparently in a perfect healthy state compared to all other non-declining forests in the center of Spain, under similar ecological conditions. Surprisingly, we discover that the healthy stands (NA-forests) were also infected by C. P. pini without visual signs of infection of any pine all through the stands. Face to this new situation, we focused our research and hypothesis in looking for the triggering factors that could be determining the expression of infection in some stands while the others remained unaffected as the NA stands included in our study. According to the forests services, they never saw NA forests affected by defoliation processes and for these reason we considered this forest as healthy conditions.
Our results about the degree of drought stress experienced during our samplings and the pine growth (lower growth in S and NS) pointed out to drier microsite conditions in these forests as a possible triggering factor. In addition, other studies had previously reported about this wide infection without symptoms. We have specified all this information in the Introduction section (L74-78) and documented with our results and discussed in the Discussion section (L366-369; L376-380; L397-402). We hope that these additional explanations and documentation can facilitate the interpretation of our results and main conclusions.
Besides, we also want to specify that phytoplasmas are unusual pathogens that cannot be isolated, and their presence can only be detected by PCR techniques. Phytoplasma quantification of infection has been tried by rtPCR or Q-PCR, two expensive technics unaffordable in our project. However, we have recommended more research about this disease in the conclusions sections (Line 478-482): “…more studies along these lines are needed to better understand how C. P. pini acts pathologically and the level of microorganism concentration to reach a threshold for the symptomatic expression”. We have also reviewed the ms. and make corrections, including the topic phytoplasma symptom and phytoplasma infection to make better understandable the text.
For example, L335-352 it is finally concluded that the change in bark terpene content between NA – NS/S trees could be a initial sign of C. P. pini infection (visible in NS trees before they decline). How can that be while NA trees are similarly infected by C. P. pini? Throughout the paragraph this remains unclear. Terpene content may reflect infection difference, but what does differentially infect NA/NS/S trees? Throughout the discussion, this remains. The smaller dbh, higher %brown needle, nb terminal buds by branch section are “negative effect of C. P. pini on S-trees”. Ok, but nothing in the results presented support that, as all your trees you studied are infected by C. P. pini and no measure of infection severity is given. You might have another cryptic pathogen (for example in roots, or virus or whatever) that induce that and that you did not detect.
Reply: Firstly, we have to specify that we detected only the presence not the amount of microorganisms infecting the pines (infection severity). These have been explained below and in other parts of this document but see lines for ex. L151; L408-410. Therefore, we do not know if trees were similarly infected. Secondly, our reasoning about the expression or the development of symptomatology associated to Phytoplasma is related to worse abiotic conditions. These worse conditions can be observed in the level of water stress developed by the trees comparatively among stands but also in the pine development since all trees were planted in a short period of time and now they have significant differences in the dbh. About the psoibility for a crytic pathogen, as we commented in previous replies, we work within a team of forest pathologists experts and also we ask local Forest Services about other possible problems associated to this die-off phenomena. We also contacted with other Forest Services in regions with the same problem (Valencia region) and we always obtained the same response. No other pathogens were observed causing this type of symptomatology.
We have modified several paragraphs in the ms. to improve our explanations related to the impact of the Phytoplasma, the limitations of our determinations and suggestions about the future with this infection in order to analyze the disease load.
The fact that 100% of the studied trees are infected, whether they are in perfectly healthy stands or in heathy trees in stands with presence of decline is interesting and little commented. It is just said that it is expected that they are “widespread”.
Reply: We have enlarged the discussion section about these results and also according to other referees comments’ (L 378-381; L399-404).
I must say that like most forest pathologist, I am not familiar with Phytoplasma. Is the 100% prevalence “normal”; this wouldn’t be unique, some root pathogens like some common Phytophthora or some leaf pathogens (oak mildew for example) might have a 100% prevalence in a sample of 5 stands. But then, there will likely be very large difference of severity between trees. Is that what is expected with Phytoplasma (difference in pathogen load in the phloem or something similar)? If so, to reach the conclusion you claim, you need to document those differences.
Reply: Again, we also want to specify that phytoplasma are unusual pathogens. We also contacted with specialist in phytoplasma disease in the Valencia region (Forest Services) where it is quite common among pines in order to certify the observed symptomatology with phytoplasma infection and share with them pathological experiences regarding the amount of impact on the trees. As previous referees have been specify in his/her comments, few studies have been done and several part of its life cycle remain unknown, including our regional forest services. For these reason it is difficult to establish some affirmations commonly used for more common pathogens like Phytophthora, Anyway, now we have specified that our analysis only take into account the presence of phytoplasma not the amount of infection (L151). In this sense we have included some paragraphs explain that other analysis out of our possibilities could help to know the impact of microorganism concentrations (L 408-410): “…future quantification of C. P. pini in host tree tissues by rt-PCR or Q-PCR could clarify if the induced level of response are correlated with microorganism concentrations”.
Some more minor comments:
L135-138. What is done is not easy to follow. 12 samples altogether and three pines per stand and disease degree. This should be 3 trees * 2 stands without symptoms +3 trees * 3 stands with symptoms *2 disease degree (healthy/declining) which would be 20 altogether. Can you clarify.
Reply: We fully agree with the reviewer that a better explanation about sampling was needed. We have modified it in order to clarify: Now it can be read from L163 to L168 :” Three trunk core samples from three pines per stand (C, C1 and C2) and disease degree (NA, NS and S) were collected in Cerros Concejiles in July 2017. This sampling was carried out at the same time than the first sampling of ecophysiological traits. After discarding some samples unable to analyze, the final number was three samples from NA trees, five samples from S trees and four samples from NS trees”.
Fig. 8. I am not sur I have understood. I expected 4 stands (P, C, C1, C2), while in P1, logging precluded to determine mortality. P is not on the figure. Is that a mistake? Should it be P/C in the X-axis instead of C? Indicate in the caption that recent logging did not enabled to determine mortality in P1.
Reply: Thank you so much for this comment, it allowed us to find a mistake in the X-axis. Now it has been solved and the sentence regarding logging has been added to the caption, as suggested. Please, find those changes in L-360-L361 and in the renamed Figure 9.